# Development of Galectin-3 Targeting Drugs for Therapeutic Applications in Various Diseases

**DOI:** 10.3390/ijms24098116

**Published:** 2023-05-01

**Authors:** Rakin Ahmed, Khairul Anam, Hafiz Ahmed

**Affiliations:** GlycoMantra Inc., Biotechnology Center, University of Maryland Baltimore County, Baltimore, MD 21250, USA

**Keywords:** galectin-3, inhibitor, cancer, NASH, fibrosis, diabetes

## Abstract

Galectin-3 (Gal3) is one of the most studied members of the galectin family that mediate various biological processes such as growth regulation, immune function, cancer metastasis, and apoptosis. Since Gal3 is pro-inflammatory, it is involved in many diseases that are associated with chronic inflammation such as cancer, organ fibrosis, and type 2 diabetes. As a multifunctional protein involved in multiple pathways of many diseases, Gal3 has generated significant interest in pharmaceutical industries. As a result, several Gal3-targeting therapeutic drugs are being developed to address unmet medical needs. Based on the PubMed search of Gal3 to date (1987–2023), here, we briefly describe its structure, carbohydrate-binding properties, endogenous ligands, and roles in various diseases. We also discuss its potential antagonists that are currently being investigated clinically or pre-clinically by the public and private companies. The updated knowledge on Gal3 function in various diseases could initiate new clinical or pre-clinical investigations to test therapeutic strategies, and some of these strategies could be successful and recognized as novel therapeutics for unmet medical needs.

## 1. Introduction

Protein-carbohydrate interactions play important roles in modulating cell-cell and cell-extracellular matrix (ECM) interactions in various biological processes of normal and disease development such as cell activation, growth regulation, cancer metastasis, and fibrogenesis. Thus, a detailed understanding of how these carbohydrate-binding proteins (lectins) interact with their partners (carbohydrate ligands) in normal and disease development is very important for the development of lectin-targeted therapeutics. Galectins are a family of at least fifteen β-galactoside-binding lectins that are involved in growth development, and the progression of various diseases such as cancer metastasis [1,2,3,4,5], organ fibrosis [6,7,8,9,10], and type 2 diabetes [11,12,13,14,15,16]. Galectins are classified as Proto, Chimera, and Tandem-repeat based on their subunit structures [17] (Figure 1). Proto-type galectins comprise one carbohydrate-recognition domain (CRD) per subunit and these are either monomer (examples: galectins-10, -11, -13, -14, and -15) or dimer (examples: galectins-1, -2, and -7). Tandem-repeat type galectins have two similar, but not identical, CRDs joined by a linker peptide (examples: galectins-4, -6, -8, -9, and -12). The chimera type galectin (galectin-3 only) also contains one CRD at the C-terminal end, but its N-terminal end is rich with proline-glycine repeats. Galectin-3 (Gal3) is a monomer, but it can form a multimer (dimer or pentamer) at higher concentrations [17].

Gal3 is one of the most studied members of the galectin family [2,18,19,20,21]. The literature search of Gal3 (using the previous name IgE binding protein) shows almost 10,000 publications at the time of this manuscript preparation. As a multifunctional protein involved in multiple pathways of many diseases, including cancer, fibrosis, and diabetes, Gal3 has generated significant interest in pharmaceutical industries [22]. This review briefly describes its structure, carbohydrate-binding properties, including the endogenous ligands, and roles in various human diseases, and it discusses the development of its potential inhibitors (antagonists) that are currently being investigated clinically or in pre-clinical animal models.

## 2. Primary and Three-Dimensional (3D) Structure of Gal3

Gal3 (previously known as Mac-2, L-29, L-31, L-34, IgE binding-protein, CBP35, and CBP30) consists of three structurally distinct domains containing a highly conserved short N-terminal domain (ND) with 12-amino acids [19], a long ND rich with proline and glycine, and a C-terminal CRD [23] (Figure 2). The short ND has been shown to have roles in Gal3 secretion and Gal3-mediated apoptosis since deletion of the short ND abrogates secretion of Gal3 [24], and mutation of the conserved Ser6 in the short ND affects Gal3’s anti-apoptotic signaling activity [25]. The long ND of Gal3 is responsible for its multimerization and positive cooperativity in carbohydrate binding [24,26]. The C-terminal CRD of Gal3, comprising approximately 130 amino acids, forms a globular structure like other galectins [19] and accommodates a pocket for carbohydrate binding [27,28,29,30]. The human Gal3 gene (*LGALS3*) approximately 17 kb long is located on locus q21–q22 of chromosome 14 [31] and contains six exons [32]—of which exons 1–3 represent the N- terminal domain, while exons 4–6 house the CRD. The open reading frame of human Gal3 mRNA is 753 bp long (NM_002306.3 for transcript variant 1).

The Gal3 CRD co-crystallized with small molecule inhibitors such as Thomsen-Friedenreich (TF) antigen (Galβ1-3GalNAcα1-*O*-Ser/Thr), lactose, TFN (TF p-nitrophenyl), and GM1 [33,34,35] shows a high similarity to the previously reported structure [36]. The Gal3 CRD adopts a typical galectin fold consisting of eleven-stranded (S1–S6 and F1–F5) antiparallel β-sheets, jointly forming a β-sandwich structure where the S1–S6 β-strands form a concave surface on which a TF antigen and other glycans are bound.

## 3. Carbohydrate-Binding Properties of Gal3 and Its Endogenous Ligands

All galectins bind β-galactoside; however, subtle differences in their carbohydrate-binding properties are observed. For example, most galectins preferentially bind N-acetyllactosamine, Galβ1,4GlcNAc (5–10 times stronger) over lactose, Galβ1,4Glc [37,38,39,40,41] and so *N*-glycans containing the N-acetyllactosamine are good ligands for most galectins. Interestingly, the interaction of Gal3 with the TF-disaccharide, Galβ1,3GalNAc found in *O*-glycans seems to be strikingly different compared to that of galectin-1 [36,37,38,39]. On isothermal titration calorimetry (ITC) assays, Gal3 interacted with the TF-antigen with a 100-fold higher affinity compared to galectin-1 [35]. The basis for these subtle differences in galectins’ carbohydrate-binding properties can be explained by their 3-D structures [35,36,42].

Gal3 has both intracellular and extracellular ligands. Like other galectins, Gal3 lacks a typical secretory signal peptide [43], and it is present in the cytosol and also in the ECM [44,45]. The β-galactoside-containing glycoproteins of the ECM and cell surface, such as laminin [46,47], fibronectin [48], CD29 [49], CD66 [50], α1β1 integrin [48], and Mac-2 binding protein, [51] are known extracellular ligands of Gal3. Among intracellular ligands of Gal3, gemin 4 [52], Bcl-2 [52], nucling [53], synexin [54], and β−catenin [55,56] are known, and Gal3 binds to these ligands via protein-carbohydrate or protein-protein interactions.

## 4. Gal3 Mediates Cell-Cell and Cell-ECM Interactions

Gal3 exerts multiple biological roles intracellularly within the nucleus or the cytoplasm, or after its secretion, at the cell surface and/or the extracellular space [3,4,17]. Gal3 binds to cell surface β-galactose-containing glycoconjugates or glycolipids, thereby regulating cell proliferation, apoptosis, cell adhesion, invasion, angiogenesis, and metastasis in normal development, as well regulating the progression of disease processes such as tumorigenesis and fibrogenesis.

### 4.1. Expression of Gal3 in Normal Growth Development

Gal3 is developmentally regulated and expressed in all types of tissues [57,58]. During mouse embryogenesis, expression of galectin-3 was first observed in the trophectoderm on day 4 and then in the notochord cells between 8.5 and 11.5 days of gestation [57]. In later stages of mouse development, expression of Gal3 was observed in the cartilage, ribs, larynx, esophagus, facial bones, a suprabasal layer of the epidermis, and endodermal lining of the bladder [19]. In the adult stage, expression of Gal3 was mostly found in epithelial cells such as small intestine [59], colon [60], cornea [61,62], kidney [63], lung [64], thymus [65], breast [66], and prostate [67]; ductal cells such as the salivary glands [68], pancreas [69], kidney [70], and eye [71]; and in intrahepatic bile ducts [72]. Among various cell types, Gal3 is expressed in fibroblasts [73], chondrocytes and osteoblasts [74], osteoclasts [75], keratinocytes [76], Schwann cells [77] and gastric mucosa [78], endothelial cells [79], and immune-related cells such as neutrophils [80], eosinophils [81], basophils and mast cells [82], Langerhans cells [76,83], dendritic cells [84], monocytes [85], and macrophages from different tissues [3,18,86,87].

### 4.2. Gal3 Is Involved in the Progression of Many Diseases

Gal3 mostly plays a pro-inflammatory role and is involved in many diseases associated with chronic inflammation, such as cancer, fibrosis, and type 2 diabetes (Figure 3).

#### 4.2.1. Gal3 Promotes Tumor Progression and Metastasis

Gal3 is expressed in various tumors [18,19,21,66,67,74,87,88]; however, the intensity of its expression depends on the type of tumor, its invasiveness, and its metastatic potential [53,54]. For example, Gal3 is highly expressed in the colon, head and neck, liver, gastric, endometrial, thyroid, skin, and breast carcinomas [55,56,66,88,89,90], while decreased expression of Gal3 is observed in the prostate [67,91], bladder [92], kidney [93], and pituitary cancers [94]. During the progression of some cancers such as colorectal [53,54,95], tongue [53,54], and prostate cancer [67,91,96], changes in cellular localization (shuttling from the nucleus to the cytoplasm) of Gal3 have been observed. For regulation of Gal3′s nuclear export, phosphorylation of Ser6 at the short ND seems important [97]. The rationale for the decreased expression of Gal3 in the early stages of pituitary and prostate cancer has been investigated by us and others, and the methylation of DNA in the Gal3 promoter has been shown to be responsible for its decreased expression [91,94,98,99].

Several studies, including ours, suggest that Gal3 can promote tumor progression and metastasis in many cancers through various mechanisms such as angiogenesis, homotypic and heterotypic aggregation, tumor-endothelial interactions, inhibiting apoptosis, and evading host immune response [2,4,18,37,88,100] (Figure 4). The role of Gal3 in neo-angiogenesis has been corroborated, as the disruption of Gal3 expression impairs angiogenesis by reducing VEGF secretion from TGFβ1-induced macrophages [101], while overexpression of Gal3 in a Gal3-deficient prostate cancer cell line, LNCaP, induced in vivo tumor growth and angiogenesis [102]. During cancer metastasis, cancer cells, after detaching from the primary tumor site, form secondary tumors by aggregating with other tumor cells in microcapillaries and extravasate at the secondary sites. During extravasation, tumor cells bind to endothelial cells, possibly through protein-carbohydrate interactions, and penetrate through the layers of endothelial cells and basement membranes. Studies suggest that Gal3 is involved in most steps of metastasis through the promotion of homotypic cell adhesion and heterotypic aggregation by binding to soluble complementary glycoconjugates [103], and interactions between tumor cells and endothelial cells, angiogenesis, and tumor metastasis [2,4,18,104]. The role of Gal3 in prostate cancer metastasis has been demonstrated, as the metastasis was blocked or prevented in an experimental metastasis assay in nude mice using Gal3 knockout PC-3 prostate cancer cells [37] and in a transgenic mouse model of aggressive metastatic prostate cancer treated with our proprietary Gal3 inhibitor, GM101 (unpublished results). The role of Gal3 in breast cancer metastasis was also investigated in an experimental liver metastasis model using human breast carcinoma BT549 cells [105]. After intrasplenic injection, only Gal3 overexpressing BT549 cells (Gal3^+^BT549), but not Gal3 null BT549 cells (Gal3^−^BT549), formed metastatic colonies in the liver, thus demonstrating Gal3′s role in the promotion of metastasis [105]. For tumor-endothelial cell interactions, Gal3 expressed in endothelium can participate in the docking of cancer cells on capillary endothelium by specifically interacting with cancer cells-associated TF-disaccharide (TFD, Galβ1,3GalNAc) present in the core I structure of mucin-type *O*-linked glycan [106,107]. In normal cells, the TFD is usually masked by sialic acid, but in malignant and premalignant epithelial cells, it is exposed or non-sialylated [106,107]. The role of Gal3 in promoting cancer cell homotypic aggregation has been appreciated through the interaction of the circulating Gal3 with TFD on the cancer-associated transmembrane mucin protein MUC1 [108,109] and, also in three-dimensional co-cultures of endothelial and epithelial cells [66].

Intracellular Gal3 can also promote tumor progression by inhibiting apoptosis of cancer cells through various mechanisms [7], such as when Cytoplasmic Gal3 binds the Bcl-2 protein and inhibits the mitochondrial-apoptotic response [110]; Gal3 promotes strong activation of PI3K (phosphoinositide 3-kinase) through interaction with activated K-Ras [111]. Gal3 transfected BT549 human breast carcinoma cells block cytochrome c release and nitric oxide-induced apoptosis [112].

Gal3 is also involved in the immune escape mechanism of cancer cells during tumor progression, as extracellular Gal3 secreted from tumor cells has been shown to induce apoptosis of cancer-infiltrating T-cells [37,49,113,114]. The presence of Gal3 in the T cells seems important, as Gal3-null T-cell lines such as Jurkat, CEM, and MOLT-4 cells are significantly more sensitive to exogenous Gal3 compared to Gal3-expressing cell lines such as SKW6.4 and H9. This has been corroborated by the observation that Gal3 transfected Jurkat cells were more resistant to apoptosis induced by anti-Fas antibodies or staurosporine than non-transfected control cells [29,115]. By secreting Gal3, cancer cells thus have acquired the ability to defend against infiltrating T-cells. For apoptosis of T cells, extracellular Gal3 binds to the CD29/CD7 complex, thereby triggering the activation of an apoptotic signaling cascade through mitochondrial cytochrome c release and activation of caspase-3 [113,116,117]. Extrinsic apoptosis may occur in two major signaling pathways, such as via death receptors Fas (apo-1/CD95) or through TRAIL (TNF-related apoptosis-inducing ligand or Apo2-L) [118,119].

In summary, Gal3 is involved in the progression of cancer metastasis and drug resistance through multiple mechanisms including the tumor microenvironment and thus the Gal3-targeting strategies could bring significant results in cancer treatment and management.

#### 4.2.2. Gal3 Is Involved in the Fibrogenesis of Various Organs

Accumulating evidence suggests that Gal3 is involved in the promotion of fibrosis of various organs such as the liver [6], lung [7], skin [8], kidney [9], and heart [10]. During tissue fibrosis, Gal3 promotes the release of pro-fibrotic factors, activation of inflammatory cells such as macrophages, the proliferation of ECM-producing cells such as fibroblasts and myofibroblasts, and tissue injury [6,7,9,120,121,122,123] (Figure 5). In this process, Gal3 is believed to cross-link with glycans of the TGF-β receptor resulting in prolonged activation of the receptor [124]. The role of Gal3 in promoting fibrogenesis was corroborated both in in vitro and in vivo experiments as the inhibition of Gal3 with carbohydrate ligands or the knockdown of Gal3 attenuated fibrosis [6,7,8,9,124,125,126,127,128,129,130,131,132,133,134]. In mouse renal fibrosis, Gal3 was shown to be overexpressed, but Gal3 deficiency inhibited renal fibrosis [9,135,136]. Gal3 has been shown to be a marker for an increased risk of heart failure, and it may play a critical role in cardiac fibrosis [10,133,137,138,139]. In lung fibrosis, TGF-β is involved in the ECM production and apoptosis of alveolar epithelial cells [140,141,142,143,144], and Gal3-TGF-β receptor binding results in prolonged receptor activation [124]. In vascular fibrosis, overexpression of Gal3 enhanced collagen I synthesis in rat vascular smooth muscle cells, [145] the inhibition of Gal3 with modified citrus pectin (MCP), and Gal3 silencing with gene-specific siRNA all resulted in blocked collagen I synthesis [145].

Liver fibrosis is driven by a heterogeneous population of hepatic myofibroblasts derived from hepatic stellate cells (HSCs, key fibrogenic cells of the liver) and portal fibroblasts [146]. The HSCs are able to phagocytose apoptotic bodies of dead hepatocytes [17,34]. Upon phagocytosis of apoptotic hepatocytes, the HSC transdifferentiates into myofibroblasts with the production of collagen I, transforming growth factor TGF-β, and reactive oxidative species. These fibroblasts facilitate hepatocyte interactions via inflammatory mediators [146] and thus, liver fibrosis is prevented in 57–79% of patients mainly by anti-inflammatory treatments [147]. Gal3 is believed to be involved in the regulation of phagocytosis-mediated HSC activation [148]. Gal3 was shown to stimulate HSC proliferation by initiating the ERK1/2 signaling pathway, while an inhibitor of Gal3 (thiodigalactoside) attenuated the effects [148]. Other galectin inhibitors, such as galactoarabino-rhamnogalacturonan or galactomannan were shown to reduce liver fibrosis in rats [149] and NASH fibrosis in C57BL/6 mice [150]. The role of Gal3 in liver fibrosis has been supported by the fact that Gal3 null mice were either resistant to the development of NASH and fibrosis [151], or attenuated inflammation and IL33-dependent fibrosis [152]. Overall, data suggest that the specific inhibition of Gal3 may represent a promising therapeutic strategy against tissue fibrosis. The Gal3-targeting strategy for fibrosis therapy is novel and significant as the Gal3 inhibitors interfere with the Gal3-TGFβ receptor binding.

#### 4.2.3. Gal3 Is Involved in Type 2 Diabetes (T2D)

T2D accounts for about 90% of all diabetes and is often associated with obesity. T2D occurs when β-islet cells in the pancreas do not produce insulin in a high enough quantity, or the cells of the body are non-reactive towards insulin. Obesity-associated inflammation and insulin resistance, mediated by macrophages and other immune cells, is a hallmark of T2D and plays a central role in metabolic syndrome [153,154]. Interestingly, Gal3 has recently been shown to cause cellular and systemic insulin resistance [11] by the Olefsky laboratory (Figure 6). They showed that the Gal3 derived from macrophages impaired glucose tolerance associated with obesity-induced T2D [11]. In animal experiments, Gal3 administered to obese mice was shown to cause insulin resistance and glucose intolerance, whereas loss of Gal3 by genetic or pharmacologic means improved insulin sensitivity [11]. To explore the mechanism of Gal3-mediated insulin resistance in T2D, they concluded that Gal3 could bind to the insulin receptor (IR), causing an inhibition of the downstream signaling. In T2D patients, the serum level of Gal3 has been associated with indices of insulin resistance [12,13,14]. The serum level of Gal3 was also found high in prediabetes [15]. Moreover, Gal3-knockout mice were found resistant to diabetogenesis, suggesting Gal3′s role in diabetogenesis [16]. Overall, data show a strong link between Gal3 and obesity-induced insulin resistance in insulin-targeted hepatocytes, adipocytes, and myocytes, and thus the specific inhibition of Gal3 may offer a potential therapeutic strategy for restoring insulin sensitivity. The Gal3-targeted strategy for T2D therapy is very significant as it would reveal a new mechanism for restoring insulin sensitivity through direct interaction with the insulin receptor.

#### 4.2.4. Gal3 in Other Diseases

Accumulating evidence suggests that Gal3 has a role in the pathophysiological mechanisms of the immune response, particularly in the recruitment, activation, and removal of neutrophils associated with asthma [130,155,156,157]. Gal3 is involved in the development of the allergic inflammatory response in atopic dermatitis [158], as analyzed in an experimental mouse model of atopic dermatitis where increased expression of Gal3 in the epidermis was observed. Overall, Gal3 has been demonstrated as a pro-inflammatory mediator of skin inflammation in atopic skin disease [158].

Gal3 is believed to have a role in chronic obstructive pulmonary disease (COPD) as the level of serum Gal3 was significantly increased in acute exacerbation of COPD compared to that in the COPD convalescence phase [159]. Gal3 may have a role in psoriasis. Gal3′s role in psoriasis was discovered unexpectedly during the NASH clinical trial with Gal3 inhibitor, GR-MD-02 where moderate to severe plaque psoriasis was effectively treated (ClinicalTrials.gov (accessed on 19 February 2023) Identifier: NCT01899859).

Several lines of evidence suggest that Gal3 could promote inflammation in rheumatoid arthritis (RA) [160]. In collagen-induced arthritic rats, increased Gal3 secretion into the plasma correlated with the disease progression [161]. The Gal3 level increased in the serum and synovial fluid of RA patients with the long-standing disease compared to that in osteoarthritis (OA) and Juvenile idiopathic arthritis (JIA) patients [162,163]. Moreover, the downregulation of Gal3 expression through therapeutic administration of Gal3 small hairpin RNA (shRNA) containing lentiviral vectors in rats with collagen-induced arthritis significantly ameliorated the disease activity [164]. Overall, data suggest that Gal3 plays a key role in the pathogenesis of RA [165], and the down-regulation of Gal3 may represent a novel therapeutic strategy for RA.

Gal3 may be involved in the pathogenesis of endometriosis, and the associated pain as increased expression of Gal3 is detected in the peritoneal fluids of women with endometriosis [166,167]. Gal3 is believed to be involved in myelin phagocytosis and Wallerian degeneration of neurons, as it can trigger neuronal apoptosis after nerve injury [168]. Gal3 was overexpressed in endometriotic foci via a nerve growth factor and could be responsible for the induction of nerve degeneration and pain [167,169].

Gal3 has been recently implicated as a potential marker of lung damage, and a predictor of poor outcomes in COVID-19 patients [170,171]. Accumulating evidence suggests that Gal3 is involved in the promotion of various viral infections, and the enhancement of pro-inflammatory cytokines such as interleukin (IL)-1, IL-6, and tumor necrosis factor (TNF)-α [170,172,173,174]. We have confirmed Gal3 binding to SARS-CoV-2 SpGp (unpublished results). Interestingly, increased levels of Gal3 are found in the blood, lung, alveolar cells, and respiratory tract mucus of COVID-19 patients [170,174,175]. The increased levels of Gal3 in the respiratory tract of COVID-19 patients could be responsible for the enhanced attachment of SARS-CoV-2 through binding to N/O-glycans of spike glycoprotein. Gal3 could induce and promote ARDS as an evolution of CSS by regulating the entire host-mediated immunologic sequela of COVID-19 and suggest that Gal3 could be a possible therapeutic target.

## 5. Commercial Development of Gal3 Inhibitors for Various Therapeutic Applications

Several Gal3 inhibitors or antagonists, either small-molecule carbohydrates or large-molecule natural products, are being developed by private and public companies for therapeutic applications in various diseases.

### 5.1. Small Molecule Gal3 Inhibitors Used in Clinical Trials

Small molecule drugs have some advantages over large molecule drugs, such as the potential to design a smaller structure with the proper characteristics (such as polar surface area, biostability) amenable for oral delivery, able to be synthesized on a large scale consistently, and well characterized and easy to navigate in pharmacokinetic studies [176]. However, synthetic molecules have some disadvantages, such as the potential to be toxic at higher doses [177]. A few studies of small molecule Gal3 inhibitors have been registered for clinical trials, mostly for lung and liver fibrosis (www.clinicaltrials.gov (accessed on 20 February 2023)) (see Table 1 for the details).

Based on the thiodigalactoside, Galecto Biotech prepared a few small molecule synthetic drugs such as TD139 (later named GB0139) and GB1211 with nanomolar affinity (2.3–25 nM) to Gal3 [177,178,179]. Galecto Biotech recently completed phase 1/2a randomized, double blind, placebo controlled clinical trial (NCT02257177) to assess the safety, tolerability, pharmacokinetics, and pharmacodynamics of inhaled TD139 (Table 1). This phase 1/2a study was evaluated in 36 healthy subjects and 24 patients with idiopathic pulmonary fibrosis (IPF) as follows: Six dose cohorts of six healthy subjects were given single doses of TD139 (0.15–50 mg), and three dose cohorts of eight patients with IPF were given once-daily doses of TD139 (0.3–10 mg) for 14 days [177]. Inhaled TD139 was found rapidly absorbed and well tolerated in healthy subjects and IPF patients. The delivery of TD139 to the target tissues was confirmed by bronchoscopies during therapy. The expression of Gal3 on alveolar macrophages was reduced in the 3 and 10 mg dose groups compared with the placebo. Moreover, inhibition of Gal3 was associated with reductions in relevant plasma biomarkers [177]. Following this successful study, a randomized, double-blind, multicenter, parallel, placebo-controlled Phase 2b clinical trial (GALACTIC-1, NCT03832946) is planned to evaluate the efficacy and safety of inhaled GB0139 in 426 IPF patients that receive 3 mg dose once daily for 52 weeks.

Another Phase 2 trial of TD139 or GB0139 (DEFINE, NCT04473053) has been initiated to evaluate its efficacy in COVID-19 patients (*n* = 200) that receive 5 mg twice daily for the first 48 h and then subsequently 5 mg once daily for the remaining 12 days or until discharge from hospital or withdrawal from the trial. The study aims to investigate the delivery potential of inhaled GB0139 in pre-ventilator patients hospitalized with COVID-19. The study also examines if the GB0139 treatment can reduce viral load and disease severity, as well as measure any changes in blood biomarkers. Although the trial is ongoing, the results from a pilot study using 40 COVID-19 patients treated with GB0139 plus SOC (standard of care) and 35 patients treated with just the SOC became available recently [180]. The GB0139 and SOC combination was well tolerated. Moreover, GB0139 plus SOC achieved clinically relevant plasma concentrations and target engagement (reduction in markers associated with inflammation, coagulation, fibrosis, and reduction in inspired oxygen over SOC alone). The results from this pilot study show the therapeutic potential of inhaled GB0139 in hospitalized patients with COVID-19.

Galecto Biotech recently developed a new synthetic Gal3 antagonist (named GB1211) for oral delivery [179] and completed a phase 1 study to assess its safety, tolerability, and pharmacokinetics in healthy adults (*n* = 78) (NCT03809052). Based on the favorable Phase 1 results, a Phase 1b of orally administered GB1211 in patients with suspected or confirmed non-alcoholic steatohepatitis (NASH) and liver fibrosis (randomized, double-blind, placebo controlled, 12-week study) (GULLIVER-1, NCT04607655) has been initiated to evaluate the safety, tolerability, pharmacokinetics (PK), and pharmacodynamics (PD). However, this trial was withdrawn for an unknown reason (clinicaltrials.gov). Instead, an open-label single and repeat dose (randomized, placebo-controlled) Phase 1 trial was initiated to assess the safety, tolerability, and PK of GB1211 in participants (*n* = 54) with hepatic impairment with Child–Pugh B and C (GULLIVER-2, NCT05009680). Moreover, additional studies (open-label and randomized, double-blind, placebo-controlled, parallel group) are being planned to investigate the safety and efficacy of GB1211 in combination with atezolizumab in patients (*n* = 102) with non-small cell lung cancer (NSCLC) (NCT05240131).

### 5.2. Large Molecule Gal3 Inhibitors in Clinical Trials

Plant derived or modified pectins are the large molecule Gal3 antagonists that have been clinically investigated in various indications (www.clinicaltrials.gov (accessed on 20 February 2023)) (Table 2). These pectins lack specificity for a particular galectin and bind Gal3 with low affinity (2.6–10 μM) [181,182,183].

#### 5.2.1. Modified Citrus Pectin (MCP)

Pectins, a family of structurally complex polysaccharides (molecular weight 50–180 kDa) found mostly on the cell walls of plants [184], are extensively used in the food industry as gelification agents. Pectins have two main fractions: linear fraction or homogalacturonan fraction (HG), and the branched fractions known as rhamnogalacturonan I (RG-I) and rhamnogalacturonan II (RG-II). Modified citrus pectins (MCP) are produced from citrus pectin by sequential alkali and acidic hydrolytic processes to enhance absorbability. Studies in cell lines and animal models suggested that MCP could be an effective anti-metastatic drug for many cancers [185,186,187,188]. The MCP was shown to inhibit in vitro tumor cell adhesion to endothelium [186] and homotypic aggregation and the formation of metastatic deposits of human breast and prostate carcinoma cells in lungs and bones in xenograft rodent models [185,187]. Recently, Gal3-targeting multifunctional core-shell glyconanoparticles based on the low molecular weight dialdehyde oligomers of citrus pectin (CPDA) have been described [188]. These CPDA-based core-shell nanoparticles have been shown to reduce homotypic cellular aggregation, tumor-endothelial cell interactions, and endothelial tube formation—the significant steps of cancer progression [188].

MCP is commercially available as a food supplement, and at least two clinical investigations of MCP have been completed so far on prostatic neoplasm (FDA-regulated, NCT01681823, Phase 2) and hypertension (non-regulated, NCT01960946). The former study of MCP on prostatic neoplasm assessed its effect on prostate-specific antigen (PSA) kinetics in biochemically relapsed prostate cancer with serial increases in PSA. Moreover, Massachusetts General Hospital (MGH) initiated a Phase 3 randomized, double-blind clinical trial of MCP on knee osteoarthritis (*n* = 50) (NCT02800629), but the recruitment status is “unknown” on clinicaltrials.gov (accessed on 20 February 2023).

#### 5.2.2. GCS-100

GCS-100 is a complex polysaccharide prepared from modified citrus pectin (MCP) and has been shown to have great potential to treat multiple myeloma cells, including those resistant to dexamethasone, melphalan, or doxorubicin by La Jolla Pharmaceuticals [189,190]. GCS-100 was found to modulate MCL-1, NOXA, and cell cycle to induce myeloma cell death [190]. Moreover, La Jolla Pharmaceuticals investigated GCS-100 on chronic kidney disease (CKD, Phases 1 and 2) and chronic lymphocytic leukemia (CLL, Phase 2). The Phase 1 study assessed the overall safety of a rising dose of GCS-100 given weekly in patients (*n* = 29) with CKD (NCT01717248). An open-label Phase 2 study (NCT00514696) to assess the safety and biological activity of GCS-100 in subjects (*n* = 12) with recurrent CLL was also completed. For this trial, the CLL patients were treated at a dose of 160 mg/m^2^ of GCS-100 in a 5-day regimen, given every 21 days. Regarding safety, GCS-100 was well-tolerated with no cases of drug-related grade 3 or 4 hematological toxicity or other serious adverse events. As for efficacy, only 25% of patients showed a partial response [176]. However, based on the favorable Phase 1 results, a placebo-controlled, randomized, single-blind Phase 2a study of GCS-100 in CKD patients (*n* = 212 from two trials, NCT01843790 and NCT02155673) was conducted. The primary endpoint was to measure a change in the estimated glomerular filtration rate (eGFR). In this study, blinded subjects were randomized 1:1:1 to receive a placebo, 1.5 or 30 mg/m^2^ GCS-100 IV weekly for 8 weeks, followed by a 5-week observational period. GCS-100, at a dose of 1.5 mg/m^2^, resulted in a statistically significant (*p* = 0.045) improvement in eGFR among CKD patients, and the effect was more pronounced (*p* = 0.029) in a subset of patients with diabetic etiology. Unfortunately, GCS-100 at a dose of 30 mg/m^2^ failed to produce a statistically significant result. The investigator attributed this lack of effect to potential off-target drug effects, as this dose was 1400-fold in excess of circulating Gal3 level on a molar basis. Despite plans to continue the GCS-100 investigation in patients with CKD (NCT02333955), La Jolla Pharmaceuticals decided to discontinue the GCS-100 trial following a discussion with the Food and Drug Administration (FDA). The FDA advised the company that they would need to demonstrate the relative contribution of each component of the complex compound (GCS-100) before advancing into late-stage development. The company’s decision to terminate the GCS-100 program was based on several conditions that include the time of effort, expense, and uncertainty of being able to adequately characterize the complex compound to the FDA’s satisfaction. It remains unclear if other polysaccharide-based compounds in development will be the subject of the same FDA requirement. Moreover, La Jolla Pharmaceuticals found that GCS-100 could be useful to treat relapsed or refractory multiple myeloma and diffuse large B cell lymphoma and planned for a Phase 1/2 trial for each indication (NCT00609817; NCT00776802) but finally terminated these trials due to a lack of funding (clinicaltrials.gov (accessed on 20 February 2023)).

#### 5.2.3. GM-CT-01 (DAVANAT)

From Guar seeds, Galectin Therapeutics developed a galactomannan polysaccharide drug GM-CT-01 consisting of a backbone of β1,4-linked mannose polymer with a side chain of α1,6-linked galactose. Studies suggested that tumor cells secreted Gal3 could bind glycosylated receptors at the surface of tumor-infiltrating lymphocytes (TIL) to form glycoprotein–galectin lattices, thereby impairing the motility and functionality of TILs [191]. GM-CT-01 was shown to correct the impaired TIL by disrupting such glycoprotein–Gal3 lattices and boosting cytokine secretion by TIL [192].

A Phase 1 open-label study (NCT00054977) of GM-CT-01 was conducted in cancer patients (*n* = 40) with advanced solid tumors to evaluate the safety and tolerability of its escalating doses in the presence or absence of 5-fluorouracil (5-FU). The study showed that varying doses of GM-CT-01 plus 5-FU were well tolerated in patients with different types of solid tumors who failed standard treatments. While this study aimed to determine the safety of GM-CT-01 as a single agent, efficacy was also assessed per RECIST criteria on patients that received GM-CT-01 plus 5-FU. Only one objective response with a median progression-free survival of 8.4 weeks was reported in this study [176]. Following this, multiple Phase 2 studies in patients with colorectal cancer (NCT0011072, NCT00388700), metastatic melanoma (NCT01723813), and cancer of the bile duct and gall bladder (NCT00386516) were planned, but these studies were later terminated or withdrawn due to financing and re-organization issues as well as some unknown reasons (www.clinicaltrials.gov (accessed on 20 February 2023)).

#### 5.2.4. GR-MD-02 (Belaceptin)

Galectin Therapeutics developed GR-MD-02 polysaccharide from apple pectin through chemical processing and modification. GR-MD-02 is a galactoarabino-rhamnogalacturonan composed of galacturonic acid, galactose, arabinose, rhamnose, and smaller amounts of other sugars. It binds to Gal1 and Gal3 but with a greater affinity to Gal3 than Gal1 [https://galectintherapeutics.com/develop-proprietary-compounds/(accessed on 20 February 2023)]. A Phase 1 study (NCT01899859) of GR-MD-02 in subjects with non-alcoholic steatohepatitis (NASH) and advanced hepatic fibrosis (*n* = 31) was completed in 2015. Based on the favorable Phase 1 results, Galectin Therapeutics initiated two Phase 2 trials—one in patients with advanced fibrosis but not cirrhosis (NASH FX, NCT02421094, *n* = 30) and the second one in patients with cirrhosis (NASH CX, NCT02462967, *n* = 162). The objective of the former trial was to evaluate multiple non-invasive liver fibrosis imaging methods in assessing the efficacy of GR-MD-02 for the treatment of NASH and advanced fibrosis (Brunt Stage 3 fibrosis). The patients received either GR-MD-02 (8 mg/kg) or a placebo through intravenous (IV) injection every other week for 16 weeks (total of nine doses). The GR-MD-02 was well-tolerated with no serious adverse events, but unfortunately, the treatment failed to show a statistically significant difference in the primary endpoint of fibrosis using LiverMultiScan (Perspectum Diagnostics) or show any difference in secondary endpoints of liver stiffness (MRE) or fibrosis (FibroScan) [176,193].

In the NASH CX trial, patients were randomized 1:1:1 to either 8 mg/kg of GR-MD-02 or 2 mg/kg of GR-MD-02, or a placebo every other week for 52 weeks and to assess efficacy, a change in hepatic venous pressure gradient (HVPG) was measured as a primary endpoint. For secondary endpoints, changes in FibroScan and the 13C-methacetin breath test as an indicator of liver metabolism were recorded. Like the NASH FX trial, GR-MD-02 in cirrhosis patients was well tolerated with only two serious adverse events. However, this trial failed to demonstrate a statistically significant difference in the primary endpoint at either dose level compared to the placebo. Interestingly, a statistically significant effect of GR-MD-02 on HVPG was observed in the subgroup of NASH cirrhosis patients without esophageal varices; but this difference in HPVG was limited to the low dose of GR-MD-02 (2 mg/kg) only (no effect was observed with the 8 mg/kg dose) [176,194]. Despite this ambiguous outcome, the company has initiated a double-blind, randomized, placebo-controlled, multicenter, Phase 2b/3 trial of GR-MD-02 in NASH cirrhosis patients without esophageal varices (NCT04365868, *n* = 1010) to evaluate its efficacy and safety.

The GR-MD-02 as a standalone drug or in combination with immune system modifying agents has been investigated in other indications such as psoriasis and metastatic melanoma. An open-label, Phase 2a pilot study with GR-MD-02 given every other week for 13 infusions, 8 mg/kg, in moderate to severe plaque psoriasis patients (*n* = 5) was completed to evaluate its safety and efficacy (NCT02407041). This study improved PASI (psoriasis area and severity index) by 51.9%, which is below the typical benchmark for continued development (an improvement in PASI 75% from baseline), yet the results are considered encouraging.

Providence Portland Medical Center sponsored studies of GR-MD-02 in combination with Ipilimumab (Yervoy, a human cytotoxic T-lymphocyte antigen 4 [CTLA-4]-blocking antibody) or Pembrolizumab (Keytruda, a programmed death receptor-1 [PD-1]-blocking antibody) in patients with melanoma and NSCLC are now being conducted (NCT02117362, NCT02575404, NCT04987996). While the open-label combination study with Keytruda is ongoing, encouraging early clinical results are being reported that demonstrate a small number of patients with advanced melanoma are achieving an objective response (two complete responses and three partial responses) [176], [Press Release, Galectin Therapeutics].

#### 5.2.5. ProLectin-M

BioOxyTran recently developed an oral drug, ProLectin-M, based on guargum galactomannan and investigated it in COVID-19 patients (*n* = 10) having mild to moderate symptoms and not requiring oxygen support in a double-blind, placebo-controlled clinical study (NCT04512027). The strategy to use the Gal3 inhibition to lower SARS-CoV-2 transmission is because of the notion that the spike protein, particularly the N-terminal domain of the spike protein, S1 showed a nearly identical structural topology to human Gal3 [171]. Patients with SARS-CoV-2 positive by RT PCR were randomized, and the effect of Prolectin-M was measured to see any reduction of the viral copy numbers over 7 days of treatment. The results of this clinical study became available recently [195]. All subjects were RT-PCR negative for SARS-CoV-2 in the Prolectin-M treatment group, but not the placebo group, from day 3 onwards. Following these results, the company has initiated two new trials, Phase 1b/2a (NCT05733780) and Phase 3 (NCT05096052) to evaluate the safety, efficacy, and pharmacokinetics of orally administered ProLectin-M, but the status is showing ‘not yet recruiting’ (clinicaltrials.gov (accessed on 25 February 2023)).

### 5.3. Preclinical Programs to Target Gal3

Several Gal3-targeted preclinical investigations using small molecules and large molecule biologics are being conducted by a few companies (Table 3). Glycomimetics is developing small molecule glycomimetic antagonists of Gal3 for cancer and fibrotic diseases and has recently declared a lead product GMI2093. Moreover, Galectin Therapeutics has initiated a discovery program to identify subcutaneous and oral forms of carbohydrates and oral small molecules. In 2018, G3 Pharmaceuticals entered an exclusive license agreement with the Henry Ford Health System in Detroit for the commercial development of Gal3 inhibitors for diastolic heart failure [AlzDiscovery.org (accessed on 25 February 2023)]. However, the status of this program is currently unknown. Based on the favorable results of the oral drug ProLectin-M on COVID-19 patients [195], BioOxyTran developed a few intravenous injectable drugs around ProLectin-M (ProLectin-I, ProLectin-F, ProLectin-V, and ProLectin-A) for various viral diseases including severe COVID-19, influenza, pulmonary fibrosis, monkeypox, ebola, and ARDS—some of them are in the Investigational New Drug (IND) stage (see Table 3).

At GlycoMantra, we are developing recombinant glycoprotein inhibitors of Gal3 (GM100 series, patent pending) based on a few modifications of its patented drug TFD100 originally isolated from edible cod [21,37]. Our scientific innovation is that we have developed several Gal3 antagonists (such as GM101, GM102 etc.) that bind Gal3 with picomolar affinities. The interactions between Gal3 and its corresponding endogenous ligands are typically strong—in the nanomolar range [37]. It is thus important to break Gal3′s natural interactions to develop a successful Gal3 targeted therapy. Our Gal3 antagonists with picomolar affinity are stronger than their natural ligands, and thus they can outcompete the interaction of Gal3 with its endogenous ligands. We demonstrated the pre-clinical efficacy of these GM100 series biologics at low doses in metastatic prostate cancer, NASH fibrosis, and type 2 diabetes (glycomantra.com (accessed on 14 March 2023)) in various animal models without any adverse side-effects. GM100 series drugs could be used as standalone drugs or in combination in various Gal3-mediated diseases.

## 6. Concluding Remarks

As shown in Figure 3, Gal3 participates in a myriad of diseases, particularly those associated with chronic inflammation, and thus it has become a validated pharmaceutical target. Subsequently, Gal3 has generated a lot of interest among pharmaceutical and biotechnology companies. As a result, strategies ranging from the design and preparation of potent synthetic small molecule antagonists (i.e., glycomimetics) to large biologics from natural sources are being employed to target Gal3 for therapeutic intervention for a variety of cancers, fibrosis, and other diseases. The heightened interest from pharmaceutical and biotechnology companies has resulted in the initiation of many new clinical (Table 1 and Table 2) or pre-clinical investigations (Table 3) to test these novel therapies, and some of these strategies could be successful and recognized as novel therapeutics for unmet medical needs. These Gal3-targeting drugs can be used as a stand-alone or in combination with the existing drugs.

## Figures and Tables

**Figure 1 ijms-24-08116-f001:**
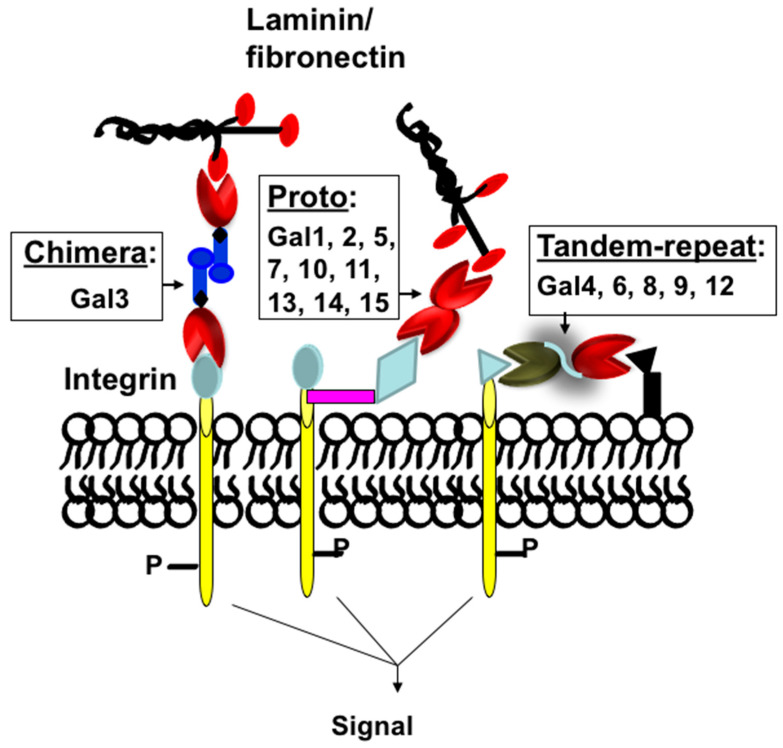
Classification of galectins. Schematic representation of proto-, chimera, and tandem-repeat type galectins. Galectins are numbered according to the order of their discovery.

**Figure 2 ijms-24-08116-f002:**
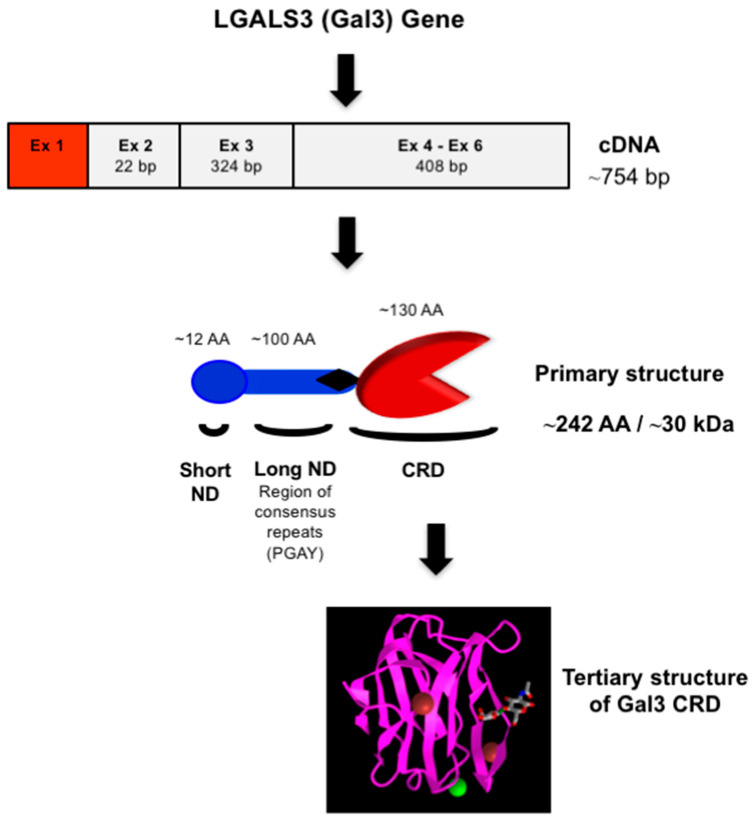
Structure of Gal3. Schematic representation of nucleotide (cDNA) and protein (primary and tertiary) structures of Gal3. The 3-D model of Gal3 CRD complexed with N-acetyllactosamine (PDB ID: 1KJL) was obtained from the NCBI (https://www.ncbi.nlm.nih.gov/Structure/pdb/1KJL (accessed on 7 November 2022)).

**Figure 3 ijms-24-08116-f003:**
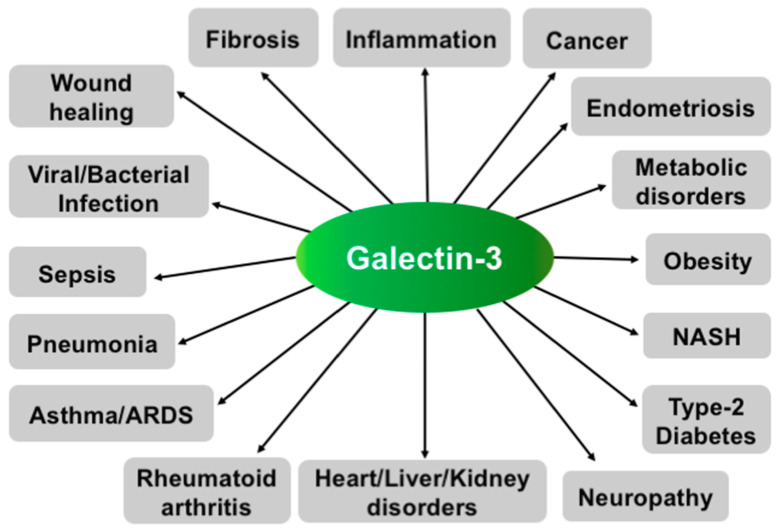
Various functions of Gal3. Schematic representation showing involvement of Gal3 in various diseases.

**Figure 4 ijms-24-08116-f004:**
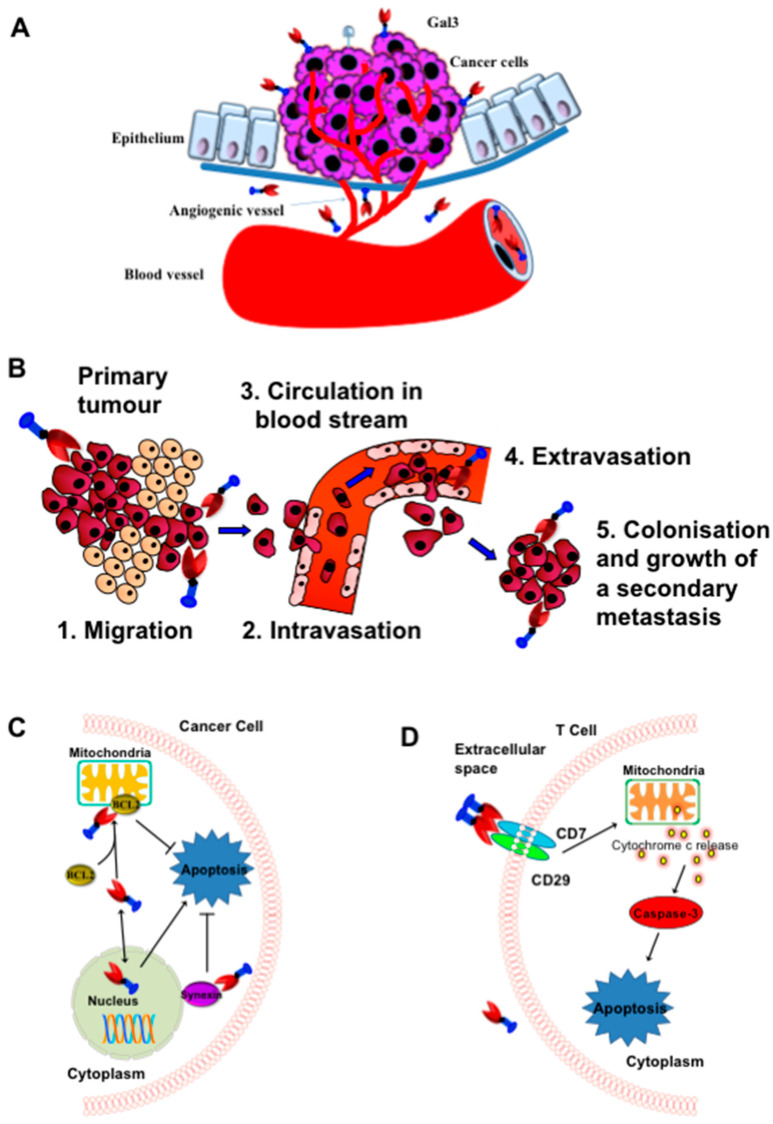
Role of Gal3 in the promotion of tumor angiogenesis, cancer metastasis, and T-cell apoptosis. (**A**) Schematic representation of Gal3 mediated tumor cell angiogenesis. (**B**) Schematic representation of Gal3 mediated tumor-endothelial cell interactions and tumor cell extravasation. (**C**) Schematic representation of anti-apoptotic function of cytoplasmic Gal3. (**D**) Schematic representation of Gal3-mediated apoptosis of T-cells. (Adapted from Ref. [21]).

**Figure 5 ijms-24-08116-f005:**
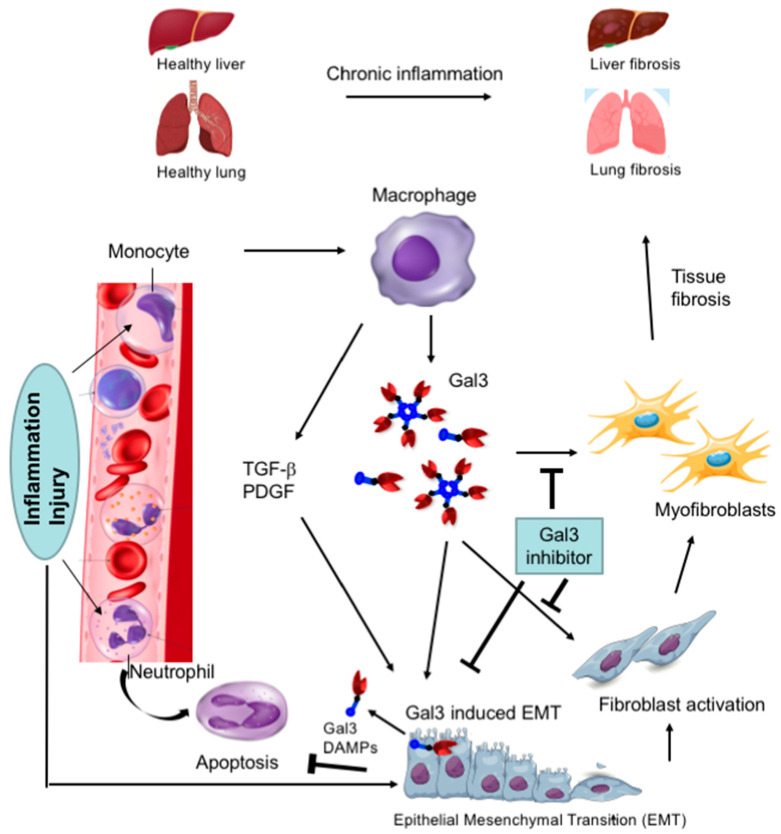
Role of Gal3 in the promotion of organ fibrosis. Schematic representation of Gal3 driven inflammatory pathways leading to organ fibrosis. TGF-β, transforming growth factor-β; PDGF, platelet-derived growth factor; DAMPS, damage-associated molecular patterns. (Adapted from Ref. [128]).

**Figure 6 ijms-24-08116-f006:**
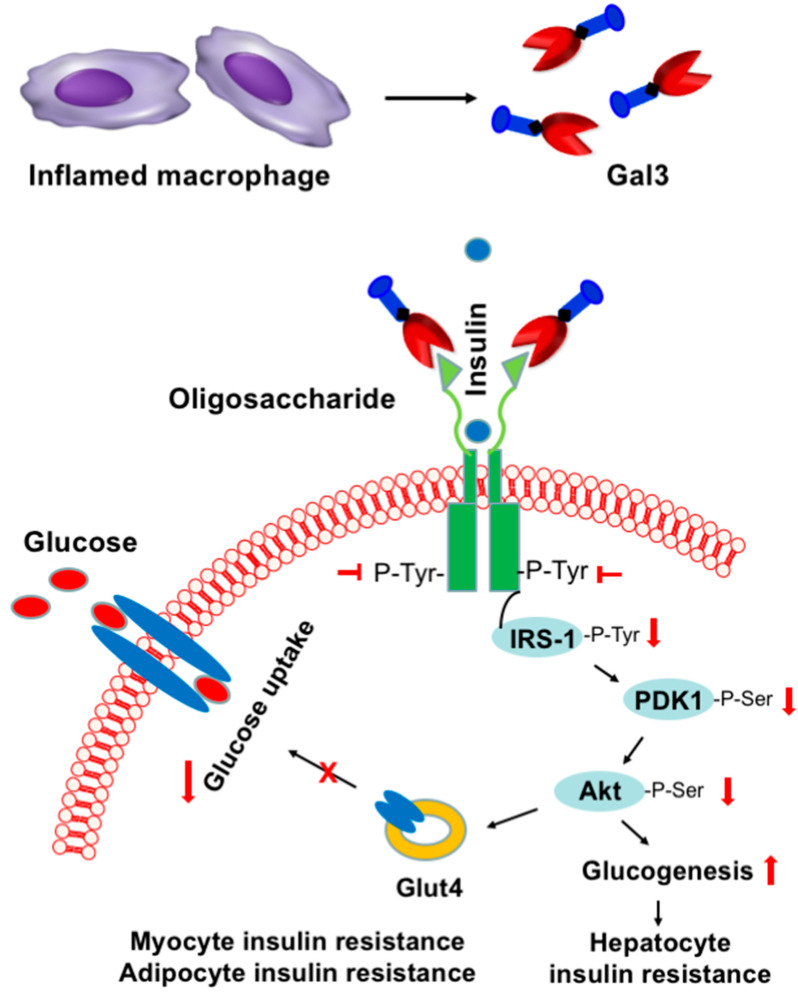
Role of Gal3 in the promotion of insulin resistance in type 2 diabetes. Schematic representation showing macrophage-secreted Gal3 from obese individuals drives insulin resistance and glucose intolerance by directly binding insulin receptor (IR) and antagonizing downstream metabolic responses. (Adapted from Ref. [11]).

**Table 1 ijms-24-08116-t001:** Clinical trials of small molecule Gal3 inhibitors.

Compound	Sponsor	Indication	Phase	Trial#	Status
TD139	Galecto Bio	IPF	1/2a	NCT02257177	Completed (*n* = 60)
GB0139	Galecto Bio	IPF	2	NCT03832946	Active, not rec (*n* = 426)
TD139	Galecto Bio	COVID-19	1,2	NCT04473053	Active, not rec (*n* = 200)
GB1211	Galecto Bio	Healthy adults	1	NCT03809052	Completed (*n* = 78)
		NASH	1b	NCT04607655	Withdrawn
		Hepatic impairment	1	NCT05009680	Active, not rec (*n* = 54)
		NSCLC	1	NCT05009680	Recruiting (*n* = 102)

Galecto Bio, Galecto Biotech; IPF, Idiopathic pulmonary fibrosis; NASH, Nonalcoholic steatohepatitis; NSCLC, Non-small cell lung cancer; not rec, not recruiting.

**Table 2 ijms-24-08116-t002:** Clinical trials of large molecule Gal3 inhibitors.

Compound	Sponsor	Indication	Phase	Trial#	Status
MCP	EcoNugenics	Prostatic neoplasm	2	NCT01681823	Completed (*n* = 60)
MCP	MGH	Osteoarthritis	3	NCT02800629	Unknown
GCS-100	La Jolla	CKD	1	NCT01717248	Completed (*n* = 29)
		CLL	2	NCT00514696	Completed (*n* = 12)
		CKD	2a	NCT01843790	Completed (*n* = 120)
			2a	NCT02155673	Completed (*n* = 92)
			2a	NCT02333955	Withdrawn
		Multiple myeloma	1/2	NCT00609817	Terminated
		B cell lymphoma	1/2	NCT00776802	Withdrawn
GM-CT-01	Gal Thera	Solid tumors	1	NCT00054977	Completed (*n* = 40)
GM-CT-01+ 5-FU	Gal Thera	Colorectal cancer	2	NCT00110721	Terminated
GM-CT-01in comb	Gal Thera	Colorectal cancer	2	NCT00388700	Withdrawn
GM-CT-01	Gal Thera	Metastatic melanoma	2	NCT01723813	Terminated
		Bile duct and Gall Bladder cancer	2	NCT00386516	Withdrawn
GR-MD-02	Gal Thera	NASH and fibrosis	1	NCT01899859	Completed (*n* = 31)
		NASH and fibrosis	2	NCT02421094	Completed (*n* = 30)
		NASH cirrhosis and Portal hypertension	2	NCT02462967	Completed (*n* = 162)
		NASH cirrhosis and Esophageal varices	2b/3	NCT04365868	Recruiting
		Psoriasis	2	NCT02407041	Completed (*n* = 5)
GR-MD-02+ Ipilumumab	Prov Med	Metastatic melanoma	1	NCT02117362	Completed (*n* = 8)
GR-MD-02+ Pembrolizumab	Prov Med	Melanoma, NSCLC	1	NCT02575404	Active, not rec
GR-MD-02+ Pembrolizumab	Prov Med	Metastatic melanoma	2	NCT04987996	Suspended
Prolectin-M	BioXyTran	COVID-19	NA	NCT04512027	Completed (*n* = 10)
		COVID-19	1b/2a	NCT05733780	Not yet recruiting
		COVID-19	3	NCT05096052	Not yet recruiting

MCP, Modified citrus pectin; MGH, Massachusetts General Hospital; CKD, Chronic kidney disease; CLL, Chronic lymphocytic leukemia; Gal Thera, Galectin Therapeutics; GM-CT-01 in comb, GM-CT-01 in combination with 5-FU, Avastin, and Leucovorin; Prov Med, Providence Portland Medical Center; NSCLC, Non-small cell lung cancer; not rec, not recruiting; NA, Not applicable.

**Table 3 ijms-24-08116-t003:** Preclinical investigations of Gal3 inhibitors (small and large molecules).

Compound	Sponsor	Indication	Status
GMI2093	Glycomimetics	Fibrosis and oncology	Achieved lead product
Belaceptin	Galectin Thera	Lung and kidney fibrosis	Preclinical
Belaceptin	Galectin Thera	Arrhythmia, Pulmonary Arterial Hypertension	Preclinical
GB1107	Galecto Bio	NSCLC	Demonstrated efficacy
MM003 (Dominant negative protein inhibitor)	MandalMed	Cardiac and Liver fibrosis	Preclinical
ProLectin-I	BioXyTran	COVID-19 (Severe), influenza	IND submission
ProLectin-F	BioXyTran	Pulmonary fibrosis, other fibrosis	IND submission
ProLectin-V	BioXyTran	Monkey pox, Conjunctivitis, Ebola	Preclinical
ProLectin-A plus Oxysense	BioXyTran	ARDS	Preclinical
GM101	GlycoMantra	Prostate, NASH fibrosis and T2D	Demonstrated efficacy
GM102	GlycoMantra	Prostate	Demonstrated efficacy

Galectin Thera, Galectin Therapeutics; Galecto Bio, Galecto Biotech; NSCLC, Non-small cell lung cancer; ARDS, Acute respiratory distress syndrome; NASH, Nonalcoholic steatohepatitis.

## Data Availability

Not Applicable.

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
