# Peer review of "Development of Galectin-3 Targeting Drugs for Therapeutic Applications in Various Diseases"

_ijms, 2023, doi:10.3390/ijms24098116_

Round 1

Reviewer 1 Report

Thank you for the opportunity to review this interesting article. The authors aim to review the structure and function of galectin-3, including its carbohydrate binding properties, endogenous ligands, and roles in various diseases. The topic in itself is important and of interest to the readers of IJMS. Overall, the manuscript is well-prepared. This is a valuable submission that I recommend for publication with a few minor changes.

Minor concerns:

1.  General proofreading for English through whole manuscript is recommended.

2. Figure 3. Various functions of galectin-3. The visibility of figure 3 can be improved.

3.   Line 122: Bladder carcinoma and melanoma can be added in the text. (Adv Cancer Res. 2023;157:157-193; J Int Med Res. 2023 Feb;51(2):3000605231153323.)

General proofreading for English through whole manuscript is recommended.

Author Response

Reviewer #1:

Thank you for the opportunity to review this interesting article. The authors aim to review the structure and function of galectin-3, including its carbohydrate-binding properties, endogenous ligands, and roles in various diseases. The topic in itself is important and of interest to the readers of IJMS. Overall, the manuscript is well-prepared. This is a valuable submission that I recommend for publication with a few minor changes.

Response: We thank the reviewer for finding our manuscript suitable for the journal.

Minor concerns:

  1. General proofreading for English throughout the whole manuscript is recommended.

Response: The proofreading and editing of the manuscript have been done by an expert.

  1. Figure 3. Various functions of galectin-3. The visibility of figure 3 can be improved.

Response: The Fig. 3 has been improved for better visibility.

  1. Line 122: Bladder carcinoma and melanoma can be added to the text. (Adv Cancer Res. 2023;157:157-193; J Int Med Res. 2023 Feb;51(2):3000605231153323.)

Response: Bladder carcinoma and melanoma were appropriately added in the text with references.

Reviewer 2 Report

This manuscript deals with "Development of Galectin-3 Targeting Drugs for Therapeutic Applications in Various Diseases" I suggest a minor correction and require a detailed clarification. A correction should be addressed by the authors as follows: The abstract is not well organized; the sentences are incomplete, and there is no sense of continuity. It would be feasible if you included the significance of the current study in the abstract. A brief description of how the authors selected information from the literature in the databases, as well as what time period they searched for, is missing. The authors should justify and expand the information on the advantages of Galectin-3 Targeting Drugs for biomedical applications. Authors should specify the main experimental conditions used based on the evidence from the literature. Where they briefly describe the most important data reported in the literature in a homogeneous manner and reinforce the relevance of Persea americana (avocado) seed husk mediated hydronium jarosite nanoparticles as novel alternatives. Authors should discuss whether the use of Galectin-3 Targeting Drugs  represents a solid alternative to existing therapeutics. Also, please discuss the use of Galectin-3 Targeting Drugs using green nanomaterials to targeting cells and mitochondria . Please add the below studies to your manuscript in the discussion section and bold your study novelties:

-Ramazanli, V. N., & Ahmadov, I. S. (2022). SYNTHESIS OF SILVER NANOPARTICLES BY USING EXTRACT OF OLIVE LEAVES. Advances in Biology & Earth Sciences Vol.7, No.3, 2022, pp.238-244 -

-Chodari, Leila, et al. "Targeting mitochondrial biogenesis with polyphenol compounds." Oxidative Medicine and Cellular Longevity 2021 (2021).

-Eftekhari, Aziz, et al. "The promising future of nano-antioxidant therapy against environmental pollutants induced-toxicities." Biomedicine & Pharmacotherapy 103 (2018): 1018-1027.

Author Response

Reviewer #2:

This manuscript deals with "Development of Galectin-3 Targeting Drugs for Therapeutic Applications in Various Diseases" I suggest a minor correction and require a detailed clarification. A correction should be addressed by the authors as follows: The abstract is not well organized; the sentences are incomplete, and there is no sense of continuity. It would be feasible if you included the significance of the current study in the abstract. A brief description of how the authors selected information from the literature in the databases, as well as what time period they searched for, is missing. The authors should justify and expand the information on the advantages of Galectin-3 Targeting Drugs for biomedical applications.

Response: We thank the reviewer for the constructive criticism of our manuscript and the helpful suggestions. The abstract is now updated with the stated significance. The time period of the literature selected is now mentioned. The advantages of Galectin-3 targeting drugs for biomedical applications are now added at the end of Sections 4.2.1, 4.2.2, and 4.2.3.

Comments: Authors should discuss whether the use of Galectin-3 Targeting Drugs represents a solid alternative to existing therapeutics.

Response: The review discusses the galectin-3 targeting drugs being investigated clinically or pre-clinically by the public and private companies. If successful, these drugs can be used as a stand-alone or in combination with the existing drugs.

Comments: Authors should specify the main experimental conditions used based on the evidence from the literature. Where they briefly describe the most important data reported in the literature in a homogeneous manner and reinforce the relevance of Persea americana (avocado) seed husk mediated hydronium jarosite nanoparticles as novel alternatives. Also, please discuss the use of Galectin-3 Targeting Drugs using green nanomaterials to targeting cells and mitochondria . Please add the below studies to your manuscript in the discussion section and bold your study novelties:

 -Ramazanli, V. N., & Ahmadov, I. S. (2022). SYNTHESIS OF SILVER NANOPARTICLES BY USING EXTRACT OF OLIVE LEAVES. Advances in Biology & Earth Sciences Vol.7, No.3, 2022, pp.238-244 -

-Chodari, Leila, et al. "Targeting mitochondrial biogenesis with polyphenol compounds." Oxidative Medicine and Cellular Longevity 2021 (2021).

-Eftekhari, Aziz, et al. "The promising future of nano-antioxidant therapy against environmental pollutants induced-toxicities." Biomedicine & Pharmacotherapy 103 (2018): 1018-1027.

Response: The review is based on the PubMed literature search. The Gal3-targeting nanoparticles based drugs, including a reference, are now discussed in the revised manuscript (please see 5.2 under MCP). However, the rest of the comments, including three suggested references, seem irrelevant to this review and so are not addressed.